# A Comprehensive Analysis of the Genomic and Expressed Repertoire of the T-Cell Receptor Beta Chain in *Equus caballus*

**DOI:** 10.3390/ani14192817

**Published:** 2024-09-29

**Authors:** Rachele Antonacci, Francesco Giannico, Roberta Moschetti, Angela Pala, Anna Caputi Jambrenghi, Serafina Massari

**Affiliations:** 1Department of Biosciences, Biotechnologies and Environment, University of Bari “Aldo Moro”, 70125 Bari, Italy; roberta.moschetti@uniba.it (R.M.); angela.pala@uniba.it (A.P.); 2Department of Veterinary Medicine, University of Bari “Aldo Moro”, 70010 Bari, Italy; francesco.giannico@uniba.it; 3Department of Soil, Plant and Food Science, University of Bari “Aldo Moro”, 70126 Bari, Italy; anna.caputijambrenghi@uniba.it; 4Department of Biological and Environmental Science and Technologies, University of Salento, 73100 Lecce, Italy; sara.massari@unisalento.it

**Keywords:** alpha–beta T cell, TRB locus, Perissodactyla, Equus, equine genome, immunogenetics, evolution

## Abstract

**Simple Summary:**

In the mammalian immune system, αβ T lymphocytes are a fundamental set of T cells capable of recognizing an extraordinary range of antigens, presented in the context of the major histocompatibility complex molecules. The function of αβ T cells is linked to the T-cell receptor, a heterodimeric cell surface protein composed of an α and β chain uniquely expressed on T cells. Unlike conventional proteins encoded by a single gene, α and β chains are encoded by a large set of noncontiguous TRA and TRB genes that are randomly assembled during a stochastic process known as V(D)J recombination, generating a large and diverse repertoire. The extent of diversification depends on the number of genes and their genomic organization at the TRA and TRB loci, which varies from species to species. In this paper, we describe the genomic organization and gene content of the horse TRB locus, based on the recently released horse genome. Compared to other mammalian species, the horse TRB locus was found to be the largest locus studied to date. A clonotype analysis of a transcriptomic dataset was also performed to assess the characteristics of the V(D)J somatic rearrangement. Overall, our data provide useful information for further functional studies in healthy and sick horses of different breeds.

**Abstract:**

In this paper, we report a comprehensive and consistent annotation of the locus encoding the β-chain of the equine T-cell receptor (TRB), as inferred from recent genome assembly using bioinformatics tools. The horse TRB locus spans approximately 1 Mb, making it the largest locus among the mammalian species studied to date, with a significantly higher number of genes related to extensive duplicative events. In the region, 136 TRBV (belonging to 29 subgroups), 2 TRBD, 13 TRBJ, and 2 TRBC genes, were identified. The general genomic organization resembles that of other mammals, with a V cluster of 135 TRBV genes located upstream of two in-tandem aligned TRBD-J-C clusters and an inverted TRBV gene at the 3′ end of the last TRBC gene. However, the horse b-chain repertoire would be affected by a high number of non-functional TRBV genes. Thus, we queried a transcriptomic dataset derived from splenic tissue of a healthy adult horse, using each TRBJ gene as a probe to analyze clonotypes encompassing the V(D)J junction. This analysis provided insights into the usage of the TRBV, TRBD, and TRBJ genes and the variability of the non-germline-encoded CDR3. Our results clearly demonstrated that the horse β-chain constitutes a complex level of variability, broadly like that described in other mammalian species.

## 1. Introduction

The horse (*Equus caballus*), a representative of the perissodactyls, is a popular farm animal. Due to its great socio-economic, biomedical, and evolutionary value, this species has played a key role throughout human history. The horse is commonly used as an animal model for human genetic diseases and, since the 19th century, has contributed to our understanding of immune response [1,2,3,4]. The current interest in equine immunology makes it necessary to investigate the genetics underlying the equine immune response, and the generation of a reference genome contributes significantly to this goal. Within adaptive immunity, T lymphocytes are key components in the context of cell-mediated response thanks to the ability of the T-cell receptor (TR) they express to perceive different peptides.

TRs are heterodimeric T-lymphocyte membrane proteins consisting of one α and β chain or one γ and δ chain. The αβ T-cell receptor is able to specifically recognize a multitude of antigenic processed peptides presented on heterologous cells by the class I or class II major histocompatibility complex (MHC) molecules [5]. γδ TRs are not MHC restricted, but they have the potential to recognize directly, without processing, diverse molecular entities, as immunoglobulins do, or presented by MHC-like proteins [6].

Each TR chain comprises a variable (V) domain in its N-terminus and a constant (C) region in the C-terminus encoded by multigene families arranged in a TR locus [7]. To generate a large repertoire of TRs capable of recognizing diverse antigens, during the development of the T lymphocytes in the thymus, the variable (V) and joining (J) genes of the TR α (TRA) and γ (TRG) loci and the V, diversity (D), and J genes of the TR β (TRB) and δ (TRD) loci undergo a somatic recombination process, with the resulting rearranged V(D)J regions encoding the V domains of the α/β and γ/δ TR chains [7].

Random trimming and addition of non-template nucleotide (N additions) at the V(D)J junctional sites greatly increase diversity and form the most variable complementarity-determining regions 3 (CDR3-IMGT) domain usually accountable for antigen recognition. Two other hypervariable loops, CDR1 (CDR1-IMGT) and CDR2 (CDR2-IMGT), are encoded by the germline V gene. CDR loops are spaced between four framework regions (FR-IMGT).

The process of somatic recombination involves the site-specific cleavage of DNA by RAG-1 and RAG-2 proteins at conserved recombination signal (RS) sequences [7]. RSs are composed of conserved heptamers and nonamers with intervening non-conserved 12- or 23-bp spacers and flank the borders of the V, D, and J genes. After transcription, the V(D)J sequence is spliced to the C gene [7].

As evident, the recombination process implies that the number of V, D, and J genes in the germline DNA is an important determinant of the extent of the primary TR repertoire, becoming the subject of study in many species. These comparative analyses, although not very useful in themselves, may reveal new insights into the multiple distinct mechanisms that may create TR diversity in vertebrates [8].

Among the four TR loci, the genomic structure of the TRB locus is the most conserved across different species of mammals, with a V gene cluster (TRBV), consisting of a number that varies from species to species, positioned upstream of tandem-aligned TRBD-J-C clusters, each composed of a single D (TRBD) gene, several J (TRBJ) genes, and one C (TRBC) gene. (https://www.imgt.org/IMGTrepertoire/, accessed on 15 July 2024; [9]) A single TRBV gene with an inverted transcriptional orientation is located at the 3′ end of the locus. In most mammalian species (https://www.imgt.org/IMGTrepertoire/, accessed on 15 July 2024; [9]), two TRBD-J-C clusters exist, while a third TRBD-J-C cluster was identified in artiodactyl species [10,11,12,13].

However, in all mammalian species, MOXD2 and EPHB6 genes border each TRB locus at the 5′ and 3′ end, respectively, whereas TRY genes are interspersed among TRBV genes, arranged in two distinct genomic positions (https://www.imgt.org/IMGTrepertoire/, accessed on 15 July 2024; [9]). 

In this paper, we provide a comprehensive genomic and expression analysis of the TRB locus in *Equus caballus* for the first time in a mammalian species belonging to the Perissodactyla. Based on the recently released horse genome, our analysis study showed that the horse TRB locus represents the largest locus identified so far in mammalian species, with 136 TRBV, 2 TRBD, 13 TRBJ, and 2 TRBC genes in approximately 900 Kb. Furthermore, the availability in public databases of a transcriptome derived from splenic tissue of a healthy adult horse allowed us to evaluate the characteristics of the V(D)J somatic rearrangements and the variability of the β-chain repertoire through a clonotype analysis. Although a significant number of germline TRBV genes are non-functional (69%), our results clearly demonstrate that the horse β-chain has a level of variability that is substantially similar to that described in other mammalian species. This is due not only to the presence of effective TRBD and TRBJ genes available for somatic recombination but also to the presence of inter-cluster and trans-cluster rearrangement processes, in addition to canonical intra-cluster recombination.

The curated data from these analyses improve our understanding of horse immunogenetics and provide insights into the evolution of these genes within mammalian species.

## 2. Materials and Methods

### 2.1. Genome Analyses

To determine the TRB locus location, the latest version of the horse EquCab3.0 genome sequence was searched using the BLAST algorithm. A sequence of 994,518 bp was retrieved directly from the reference sequence NC_009147 (*Equus caballus* chromosome 4 genomic scaffold), available at NCBI from 95080400 to 96074910 positions. Particularly, the analyzed region comprises the monooxygenase, dopamine-beta-hydroxylase-like 2 (MOXD2), and ephrin type-B receptor 6 (EPHB6) genes, flanking, respectively, the 5′ and 3′ ends of the TRB locus.

The human genomic sequence was used against the *Equus caballus* genome sequence to identify, based on homology by the BLAST program, the corresponding genomic TRBV, TRBD, TRBJ, and TRBC genes. The beginning and end of each coding exon were identified with accuracy by the presence of splice sites or flanking recombination signal (RS) sequences of the TRBV, TRBD, and TRBJ genes. The location of the TRB genes is provided in Appendix A. The sequence comparison has also allowed the identification and characterization of the horse protease genes (TRY) within the TRB locus. The locations of the TRY genes are also provided in Appendix A.

Computational analysis of the horse TRB locus was conducted using the following programs: RepeatMasker for the identification of genome-wide repeats and low complexity regions [Smit, A.F.A. Hubley, R. Green, P. RepeatMasker open-4.0. at http://www.repeatmasker.org (accessed on 11 March 2024)] and PipMaker [14] for the alignment of the horse TRB sequence against itself.

### 2.2. Classification of the Horse TR Genes

Considering the percentage of nucleotide identity of the genes with respect to humans and based on the genomic position within the locus, each horse TRB gene was classified, and the nomenclature and functionality was established according to IMGT at http://www.imgt.org/IMGTScientificChart/SequenceDescription/IMGTfunctionality.html (accessed on 4 March 2024) [15] (see Appendix A).

The TRBV genes were assigned to 29 different subgroups based on the percentage of nucleotide identity by using the Clustal Omega alignment tool, which is available at EMBL-EBI website (http://www.ebi.ac.uk/, (accessed on 4 March 2024 [16]), adopting the criterion that sequences with a nucleotide identity of more than 75% in the coding region of a TRV gene (i.e., L-PART1+V-EXON) belong to the same subgroup [17].

Subsequently, the name of each subgroup was assigned based on the comparative and phylogenetic analysis with the human TRBV subgroups according to the IMGT nomenclature [18]. Thus, the horse TRBV genes were named by the number of the assigned subgroup followed by a hyphen and a number corresponding to their genomic position within the locus when the TRBV subgroups consisted of more than one member gene.

The TRBD, TRBJ, and TRBC genes were annotated according to their similarity with humans (https://www.imgt.org/IMGTrepertoire/LocusGenes/#h1_6, accessed on 4 March 2024). Each TRBJ gene was designated by a number in accordance with the name of the belonging D-J-C cluster followed by a hyphen and a number corresponding to their position within the cluster. They were all predicted to be functional (Appendix A). 

### 2.3. Phylogenetic Analysis

The human TRBV gene sequences used for the phylogenetic analysis were retrieved from the IMGT^®^ (IMGT Repertoire (IG and TR), https://www.imgt.org/IMGTrepertoire/. Locus and genes, accessed on 4 March 2024), IMGT/GENE-DB [19]. For the phylogenetic analysis of the horse, we combined the nucleotide sequences of all V-REGION of the TRBV genes of the horse with the corresponding gene sequences of humans. All functional genes, ORFs, and pseudogenes (excepted for the TRBV1, TRBVA, TRBVB, and TRBVC pseudogenes) were selected.

Multiple alignments of the gene sequences under analysis were carried out with the MUSCLE program [20]. The evolutionary analyses were conducted in MEGA X [21,22]. We used the neighbor-joining (NJ) method to reconstruct the phylogenetic tree [23]. The evolutionary distances were computed using the maximum composite likelihood method [24] and are in the units of the number of base differences per site.

### 2.4. Horse Transcriptome Analysis

An RNA-seq transcriptomic dataset derived from the spleen of an adult healthy horse and available at the NCBI Sequencing Reader Archive (SRA, ID: ERX2600993) was examined to identify distinct TRB clonotypes. All the 13 horse TRBJ germline gene sequences were used as a probe to analyze the transcriptome data and to create datasets distinct for each TRBJ gene, considering only sequences with a percentage of nucleotide identity from 98 to 100%. The resulting reads of each dataset were then translated, and only the unique in-frame sequence with recognizable TRBJ and TRBC genes were analyzed in detail. We obtained an output comprising continuous sequences of a length of 120 bp, each containing the 3′ part of the TRBV region, the D region with any no-template bases, the TRBJ region, and the 5′ part of the TRBC region. Therefore, all reads included the V-beta CDR3-IMGT, defined as the amino acid stretch 105–117, starting at the codon after the last cysteine (2nd-CYS 104) of the TRBV gene and ending at the amino acid before the phenylalanine (J-PHE 118) in the conserved motif FGXG of TRBJ genes [15]. 

Since the TRBD genes undergo substantial base deletion and overall transformation during V-D-J recombination, the determination of TRBD genes in CDR3 is usually complex. With respect to the two TRBD genomic sequences, we chose to consider the nt located in the CDR3 region as belonging to a TRBD sequence if they constituted a stretch of at least six consecutive nt or 2 AA (50% of the length of the TRBD1 germline gene) corresponding to the TRBD1 or TRBD2 germline sequences. Unambiguous detection of TRBD bases allows for precise identification of TRBD segment boundaries and characterization of non-templated bases at the V-D and D-J junction.

## 3. Results

### 3.1. General Organization of the Horse TRB Locus

To determine the genomic organization of the horse TRB locus, we retrieved from the whole chromosome 4 contig (NC_009147) a continuous sequence between the MOXD2 and the EPHB6 genes, which, flanking all mammalian TRB loci studied to date, represent the IMGT 5′ and the 3′ borne, respectively (IMGT, https://www.imgt.org/IMGTrepertoire/LocusGenes/bornes/bornesTRB.html, accessed on 15 July 2024). Within this region of 930,869 bp, we identified and annotated all the horse TRB genes using the human sequence as reference. The first TRB gene (TRBV1) is located 4.5 Kb downstream of the IMGT 5′ borne (MOXD2 gene), while the last TRB gene (TRBV30) lies approximately 42 Kb upstream of the 3′ borne (EPHB6 gene). 

Overall, we identified 136 TRBV genes, 2 TRBD, 13 TRBJ, and 2 TRBC genes. A total of 135 TRBV genes are in a region of about 900 Kb positioned upstream of two TRBD-J-C clusters, followed by a single TRBV with inverted transcriptional orientation. One TRBD, six TRBJs, and one TRBC gene compose the TRBD-J-C cluster 1, while one TRBD, seven TRBJs, and one TRBC gene are found in the TRBD-J-C cluster2. TRBD-J-C cluster 1 spans approximately 6.7 Kb, with the six TRBJ genes occupying approximately 2.1 Kb. TRBD-J-C cluster 2 spans 6.5 Kb, of which the seven TRBJs occupy 1.1 Kb (Figure 1). 

The classification, position, and predicted functionality of all horse TRB genes are reported in Appendix A.

Additionally, we identified and annotated the tripsin-like serine protease (TRY) genes that are typically interspersed among the mammalian TRB genes (https://www.imgt.org/IMGTrepertoire/, accessed on 4 March 2024; [9]). Six TRY genes are located in 80 Kb, between the first and the second TRBV genes, while a single TRY gene is present upstream of the TRBD-J-C cluster 1 (Figure 1). The TRY genes were named with consecutive numbers based on their genomic location within the TRB locus. The position and predicted functionality of all unrelated horse TRB genes are also reported in Appendix A.

### 3.2. Sequence Analysis of the TRBV Genes and Classification

The annotated TRBV genes were grouped into 29 distinct subgroups according to the criterion that sequences with nucleotide identity above 75% belong to the same subgroup [17]. However, the membership of some genes to their own subgroup was then confirmed by phylogenetic analysis (see below), as the nucleotide identity with some but not all other members was found to be less than 75%. 

A total of 5 TRBV subgroups consist of 1 gene, while the remaining 24 contain at least 2 member genes, with a massive expansion of the TRBV5 (20 genes) and the TRBV7 (11 genes) (Table 1). Out of the 136 TRBV genes, 47 (approximately 35%) are predicted to be functional genes as defined by the IMGT rules (see Section 2), and 89 (65%) are not functional (84 pseudogenes and 5 ORFs). 

The high number of non-functional TRBV genes significantly reduces the possible horse beta-chain repertoire, with 10 subgroups containing only non-functional genes, or with subgroups such as the TRBV5, where only six of twenty were maintained as functional genes. On the contrary, five multi-member subgroups (TRBV3, TRBV6, TRBV21, TRBV27, and TRBV) are formed by only functional TRBV genes. Thus, only 19 of the 29 subgroups would contribute to generating the horse’s beta-chain repertoire.

To validate the membership of the horse TRBV genes to the subgroups and classify each TRBV subgroup, the evolutionary relationship of these genes was investigated by comparing all horse gene sequences with all corresponding human gene sequences (except for the TRBV1, TRBVA, TRBVB, and TRBVC pseudogenes). For this purpose, we combined the nucleotide sequences of the V-REGION of all selected TRBV genes in the same alignment and constructed an unrooted phylogenetic tree by the NJ method [23] (Figure 2). The tree shows a clear phylogenetic clustering of each horse TRBV subgroup (except for the TRBV1) with a corresponding human one. Therefore, based on the monophyletic groupings, each horse TRBV subgroup was named as the orthologous human one in accordance with IMGT rules [18]. The exception is the horse TRBV1 gene, which lacks the human counterpart in the tree. To assign membership to the horse TRBV1, we took into consideration the bovine and mouse TRBV1 genes, because both are located within the TRB locus in a region of more probable orthology. We then aligned the nucleotide sequences of horse, mouse, and bovine TRBV1, as well as human TRBV1, for comparison. As shown in Appendix A, the horse TRBV1 gene retains higher homology to the bovine and mouse TRBV1 genes that it does to the human one. The protein structure also confirms the orthology of the horse, mouse, and bovine genes (Appendix A). However, as in mice and cattle, we have retained the name TRBV1, being the first gene also in the horse locus.

Interestingly, all but one of the human TRBV subgroups, the TRBV9, are found in the horse genome, indicating that the emergence of distinct subgroups occurred before the divergence of the two species. 

Appendix A displays the structure of the potential functional germline TRBV genes, ORFs, and in-frame pseudogenes, as inferred from the nucleotide sequences. The amino acid sequences are aligned according to IMGT unique numbering for the V-REGION [25] to maximize the percentage of identity. Characteristic structural features (i.e., CDR-IMGT and FR-IMGT) of the genes for each TRBV subgroup as well as the amino acid sequence heterogeneity between and within subgroups are evident. It is notable that three germline TRBV genes (TRBV6-1, TRBV6-3, and TRBV25) have as a unique defect an in-frame STOP-CODON at their region end (3′ last codon for V-REGION based on IMGT) [25]. These genes are classified as functional since this STOP-CODON may very frequently disappear during V-(D)-J rearrangements owing to the mechanisms of junctional and/or N-diversity, and this results in a potential productive rearranged transcript.

### 3.3. Classification and Structure of the TRBD, TRBJ, and TRBC Genes

As in humans and mostly mammalian species (https://www.imgt.org/IMGTrepertoire/, accessed on 4 March 2024; [26,27]), the horse TRBD, TRBJ, and TRBC genes are distributed within two in tandem TRBD-J-C clusters positioned at the 3′ end of the TRB locus, with a number corresponding to their position from 5′ to 3′ (Figure 1). 

The nucleotide and deduced amino acid sequences of the two TRBD genes are showed in Appendix A. They consist of 12 (TRBD1) and 17 (TRBD2) G-rich stretches that can be productively read through their three coding phases and encode till four glycine residues, depending on the phase. The RSs that flank the 5′ and 3′ sides of the coding region are well conserved with respect to the consensus.

Appendix A reports the nucleotides and deduced amino acid sequences of all the TRBJ genes. They were classified based on the international nomenclature (IMGT, http://www.imgt.org, accessed on 04 March 2024; [19,28]) according to the subsequent TRBC gene and numbered in agreement with the genomic position within the respective cluster. Their length is between 47 bp and 53 bp and all conserve the canonical FGXG amino acid motif, whose presence defines the functionality of the J genes. Each TRBJ is flanked by a 12 RS at the 5′ end and by a donor splice site at the 3′ end. All the RSs are well conserved with respect to the consensus.

Like in all mammalian species (https://www.imgt.org/IMGTrepertoire/Proteins/, accessed on 4 March 2024), the horse TRBC genes are composed of four exons and three introns. They are nearly identical and encode a similar protein of 177 amino acids (Appendix A). Only three mismatches differentiate the nucleotide sequence of the coding region of the two genes, resulting in three amino acid (AA) changes, one in the C-domain and two in cytoplasmic region. The C-domain encoded by EX1 is 129 AA in length. The C region also comprises a connecting region (CO) of 21 AA (encoded by EX2 and the 5′ part of EX3) with a cysteine involved in interchain disulphide binding, a transmembrane region (TM) of 22 AA (encoded by the 3′ part of EX3 and the first codon of EX4), and a cytoplasmic region (CY) of 5 AA (encoded by EX4).

Minimal length differences were detected in the first (459 bp and 455 bp for TRBC1 and TRBC2 gene, respectively), the second (141 bp and 140 pb for TRBC1 and TRBC2 gene, respectively), and the third introns (311 bp and 296 pb long in the TRBC1 and TRBC2 genes, respectively). The percentage of nucleotide identity is greater than 98% between the first introns and between the second introns, while it is significantly reduced (31%) between the third introns. Likewise, the 3′UTR differs extensively both in length (222 bp for TRBC1 and 174 bp for TRBC2) and in composition (42% of nucleotide identity). The similarities and differences found between the two horse TRBC genes are a structural and distinctive feature of the TRBC genes conserved in many other mammalian species [10,26,27], indicating a likely distinctive role between the two genes.

### 3.4. Genomic Architecture of the TRBV Cluster

To investigate on the genomic architecture of the horse TRBV cluster, we first analyzed the level of GC content and the composition of the repeated sequences interspersed in the region. For this purpose, we carried out the analysis using the Repeat Masker program on the genomic sequence of approximately 900 Kb comprising genes from TRBV1 to TRBV29. The results are summarized in Appendix A. The GC content is 41.62%. The density of total interspersed repeats is 30.35%. LINEs, predominantly of the LINE1 family, are the most abundant repeat elements (19.77%). Conversely, SINE are poorly represented (1.66%).

The masked sequence was then aligned against itself with the PipMaker program to identify, through identity lines, the model of duplications that have generated the expansion of the TRBV gene subgroups (Figure 3). The dot matrix confirms the high level of nucleotide identity between TRBV genes as indicated by dots that correspond with gene location. Furthermore, we observed, in addition to the perfect main diagonal line for the match of each base with itself, parallel lines identifying duplicate portions of the sequence, which align with multiple regions within the sequence itself. 

The parallel lines are of different sizes, indicating that individual genes (the smaller lines) but also groups of contiguous genes (the larger lines) were involved during the duplication events. Different duplication units can be distinguished involving different groups of contiguous genes, which alternate within the locus. A first group consists of the genes from the TRBV2 to the TRBV8 subgroup which, although with various gene arrangements, has been duplicated several times in the locus (left-hand part of Figure 3), generating the most numerous subgroups, such as the TRBV5 (20 genes) and the TRBV7 (11 genes). A second duplicated gene block involved from the TRBV10 to TRBV13 subgroups (central part of Figure 3), including the multimember TRBV12 subgroup (7 genes). Furthermore, seven duplicate gene blocks of different sizes involve the subsequent gene subgroups, from the TRBV14 to the TRBV20 subgroup (right-hand part of Figure 3), the latter consisting of eight member genes.

Looking in more detail, it is possible to recognize longer parallel lines in the matrix made up of the union of the three different units, which form a higher-order repetitive unit in tandem duplicated.

Furthermore, small parallel lines are evident at the 3′ end of the V-cluster region (Figure 3), where the homology units corresponding to the TRBV21-TRBV22-TRBV23 and TRBV26-TRBV27 gene blocks are duplicated. Tandem duplications of the TRBV28 subgroup genes are also evident.

The PipMaker program also allows the expression of the sequence alignment as a percentage identity plot (pip) [14], as shown in Appendix A, where the position of all genes identified in the region is shown, as well as the position and orientation of all repetitive sequences. In this regard, among the interspersed repeats, we note the high percentage of LINEs located along the entire sequence, in particular between the different duplication units, suggesting their possible role in duplication events.

### 3.5. Clonotype Analysis

Genomic information of all the TRB genes and their organization allowed us to evaluate the characteristics of the V-D-J rearrangement through the CDR3 analysis of the horse beta-chain transcripts available in public databases. Since CDR3 comprises the region of juxtaposition between the rearranged TRBV, TRBD, and TRBJ genes, our analysis also provided information on the diversity of the horse beta-chain repertoire.

Essentially, we queried a transcriptome dataset derived from splenic tissue of a healthy adult horse using the sequences of each previously annotated horse TRBJ germline gene. Transcript sequences of 120 bp in length were recovered, and all were translated and screened for the presence of an open reading frame and a recognizable CDR3 sequence, with in-frame TRBJ and TRBC sequences. 

In total, 212 different clonotypes were selected. They are reported in Appendix A, grouped based on the TRBJ genes. All the germline TRBJs, except TRBJ1-3, were found in the transcripts, and, in all cases, they perfectly matched the corresponding germline genes. However, the number of transcripts for each TRBJ gene varies considerably, from 4 reads for the TRBJ1-4 and TRBJ1-5 genes to 42 for the TRBJ2-1 gene. This difference could be related to the level of participation of each TRBJ gene in the rearrangement. If so, the TRBJ2 gene cluster seems to be preferentially used (148/212 clonotypes, approximately 70%), suggesting that it is more functionally relevant for the production of horse beta-chain transcript (Table 2). 

In addition, a slight over-representation in the use of TRBJ2-1 (42/148 clonotypes, 28.3%) and TRBJ2-4 (37/148 clonotypes, 25%) can be observed.

With our experimental approach (see Section 2), a TRBD gene was identified in 134 out of 212 clonotypes, of which 53 belonging to the TRBD1 and 81 to the TRBD2 gene. Overall, 78 sequences do not have a recognizable TRBD gene. Intra-cluster D-J rearrangements represent a consistent portion of the repertoire (109/134 clonotypes, 81.3%), with 32 TRBD1-TRBJ1 and 77 TRBD2-TRBJ2 rearrangements, while 21 clonotypes (15.6%) can be interpreted as direct 5′-to-3′ joining across clusters (inter-cluster rearrangements) (in red in Appendix A). Interestingly, we also observed four TRBD2-TRBJ1 joining (in blue in Appendix A). Since the D-J-C cluster 2 is located downstream of the D-J-C cluster 1 within the TRB locus, these junctions may be explained by chromosomal inversion or, with more probability, by trans-rearrangement occurring during TRB locus recombination, as already described [26,29]. 

In the clonotypes, only the last portion of TRBV genes, involved in the second step of somatic recombination, is available to recognize and classify them. This fact, together with the often-high sequence homology between some TRBV genes and the replacement of deleted TRBV ends with non-templated bases, has made it impossible to assign a single unique TRBV gene to each clonotype. We were unable to assign a TRBV gene in 51 out of 212 clonotypes. For the remaining 161 sequences, in 93 clonotypes, the V portion was unambiguously assigned to the corresponding TRBV member genes, while in 68 clonotypes, we could only establish its membership to a subgroup (Table 3).

A substantial portion of the 161 clonotypes is represented by the TRBV20 (34 reads), TRBV5 (26 reads), TRBV21 (21 reads), TRBV28 (17 reads), TRBV14 (14 reads), and TRBV12 (11 reads) subgroups; in particular, within the TRBV20 subgroup, the TRBV20-8 gene is the most abundant, with 26 clonotypes. Among the 161 mRNA clones, we also found two members of the TRBV8 subgroup, TRBV8-3 and TRBV8-5 genes, and the TRBV5-4 and TRBV20-8 genes, all identified in the assembly as non-functional (Appendix A, Figure 1). It is possible that these clonotypes represent the transcription of an initial non-productive rearrangement. Alternatively, one can hypothesize that functional alleles of these genes exist, or, more likely, that there are sequencing errors in the genomic assembly. Further analysis will be necessary to resolve these ambiguities.

Conversely, among the functional TRBV subgroups, we did not find genes of the TRBV1, TRBV10, TRBV13, and TRBV16 subgroups. 

A close inspection of the CDR3 sequence (underlined in Appendix A) reveals that it is heterogeneous in amino acid composition. The mean length of the CDR3 loop was approximately 12.60 AA (range 9–18), with no differences among TRBJ genes (Table 2). For comparison, the human, dog, and pig peripheral blood CDR3β loops are approximately 12.7, 12.9, and 12.2 residues long, respectively [12,26,30].

## 4. Discussion

In modern times, interest in the species *Equus caballus* continues to promote genetic studies on this species, and, undoubtedly, the generation of the reference genome assembly holds the promise to further improve our knowledge of it. 

Horse genome analysis is indeed a very active area of research, aimed at identifying the genes responsible for various hereditary traits associated with fertility, athletic performance, disease resistance, etc., and at delineating the genetic makeup of individual horse breeds and populations [1,2,3,4]. The investigation also extends to the equine immune response, and, in this context, it begins to identify specific situations in which the horse may provide a unique immunological model for human diseases [1,2,3,4].

The present study, by defining the organization of the horse TRB locus, contributes to a better understanding of the repertoire of horse TRB genes and their evolution.

As expected, the general genomic organization of the horse TRB locus is similar to that of most other mammalian species in which it has been defined, with a V cluster positioned upstream of two tandem-aligned TRBD-J-C clusters, although a substantial difference is its extent, making it the largest TRB locus identified to date. Notably, the same feature was found in the analysis of the previously annotated horse TRG locus [31]. Much of the large horse TRB area (900 KB) is occupied by the V cluster (135 TRBV genes), while the two TRBD-J-C clusters span approximately 6.5 Kb each. The size of the V cluster can be related to the extent and complexity of gene duplications that have occurred in this region, which led to the emergence of numerous multi-member TRBV subgroups. In fact, only 5 of the 29 TRBV subgroups are represented by a single member gene. In comparison, in humans, the majority of the TRBV subgroups consists of a single member gene. It is noteworthy that within large subgroups, such as the TRBV5 subgroup, there is greater diversification among some member genes (nucleotide identity is less than 75%), which would indicate the need for evolution to find new functions. These genes have nevertheless been assigned to their own subgroup by phylogenetic analysis. A similar situation was found in the sheep TRD locus among the TRDV1 genes whose subgroup is represented by over 40 member genes [32]. 

Since our phylogenetic analysis revealed the clustering of each horse subgroup with a corresponding human one, except for the TRBV9 subgroup, which is missing in the horse, we can establish that gene duplication within each subgroup—rather than the emergence of distinct subgroups—is the major mode of evolution of horse TRBV genes.

It is noteworthy that gene duplications generated a horse TRBV germline repertoire consisting of a higher percentage of pseudogenes (66.4%) than the human one (26.5%). However, the total number of functional genes is identical (47), although differences in the profile of the germline functional genes can be observed, such as the absence of functional TRBV1 and TRBV21 genes in humans. 

Dot plot analyses of the TRBV region reinforce the conclusion that the V cluster arose through a series of complex duplications and suggest that these duplication events exhibit different patterns, with genes duplicating individually or genes duplicating together giving rise to extensive duplication units. The overall view of the matrix highlights how each large gene unit has duplicated itself along the entire cluster V, interspersing itself with the others. 

In another mammalian species in which the architecture of the TRB locus has been analyzed, duplications have generally involved one or a few portions (usually at the 5′) of the V cluster, and the gene units were mostly tandemly duplicated [12,14,19,28]. 

It is possible that these duplication events were mediated by repetitive sequences. Within the horse TRB locus, the percentage of repetitive elements is 30.35%, with LINEs being predominant over SINEs (19.77% versus 1.66%). The percentage of LINEs is the highest, and that of SINEs is the lowest found among various previously analyzed TRB loci [12,13,26,27]. For instance, in the human TRB, LINEs are 16.86%, and SINE are 6.62%; in dog, 11.98% of repeat sequence are LINEs, and 7.19% are SINEs; in pig and rabbit TRB loci, LINEs are 14.62% and 10.81%, respectively, even if the most abundant repeat elements are SINEs (15.65% and 14.18%, respectively).

Therefore, LINEs may have played a key role in the evolution of the horse TRB locus. Indeed, if we look closely at the pip of Appendix A, we can observe that LINEs are distributed along the entire locus and are particularly localized between duplicated blocks, which strongly suggests their contribution to the architecture of the locus. 

To explore the functional implication of the genomic characteristics of the horse TRB locus, we analyzed the V-D and D-J junctions, including CDR3, of transcripts retrieved from a public splenic RNA library. Although the number of unique clones analyzed was not very high and referred to a single tissue of a single adult animal, and although TRBV or TRBD genes could not be identified in all clones, some considerations can be made. With the exception of TRBV1, TRBV10, TRBV13, and TRBV16, all TRBV subgroups containing functional genes were found in the clonotype collection, and the TRBV usage ranged from 0.006% of TRBV29 to 21.1% of TRBV20 or 16.1% of TRBV5, considering only the subset of clonotypes where TRBV subgroup assignment was unambiguous. TRBV21 (13.0%) and TRBV28 (10.5%) subgroup genes were also frequently found.

For comparison, TRBV20-1, TRBV5-1, TRBV29, and TRBV28 represent the top four most expressed TRBV genes in human peripheral blood leukocytes collected from 550 individuals [33]. 

By CDR3 analysis, we also determined that 39.5% of recognizable TRBD clonotypes are TRBD1, and 60.5% are TRBD2. The preferential usage of TRBD2 results from intra-cluster rearrangements but also from an inter-cluster rearrangement and trans-rearrangements. This latter type of rearrangement was also found in dogs and sheep [19,31]. For comparison, an almost identical use of the two TRBD genes was observed in human peripheral blood leukocytes, with a slight prevalence for TRBD2 (50.1%) compared to TRBD1 (49.9%) [ref]. Conversely, in dog peripheral blood leukocytes, the TRBD1 gene appears more represented (71.4%) than TRBD2 [26]. However, variations in the TRBD usage may also be due to the difficulty in recognizing these genes in the transcripts.

Finally, CDR3 analysis revealed that the TRBJ2 cluster appears to be preferentially used (about 70%). It is possible that the preferential usage of the TRBJ2 set may depend on the number of J genes concentrated in a small genomic region, if multiple 12-RSs are important to increase the local recruitment of the RAG proteins [34]. In this regard, it is notable that the TRBJ1 gene set of all the species (IMGT, http://www.imgt.org, accessed on 04 March 2024; [26,27]), including the horse, is located in approximately 2.1 Kb, while the TRBJ2 genes are grouped in approximately 1 Kb. Therefore, TRBJ2s seem to be crucial for the production of beta-chain transcripts. Alternatively, the frequency variation may depend on the tissue analyzed. However, regardless of the genes involved in somatic recombination, the conservation of CDR3 length across species [12,26,30] indicates that it is a TR beta-chain feature essential for T-cell function.

Furthermore, it must be said that the number of clonotypes examined is relatively small, so this can be considered a preliminary investigation of the expressed TRB gene repertoire in *Equus caballus*. Future studies should expand the same analysis, incorporating larger datasets to consolidate the observations made in this study.

## 5. Conclusions

The annotation of the horse TRB locus is an important achievement that follows that made for the TRG locus [31]. Taken together, these findings show that both the horse TRB and TRG locus are characterized by a large size with a concomitant extensive TRV gene expansion that determines the germline TRB and TRG repertoires, suggesting that a strong evolutionary pressure has driven the enlargement of the TR regions to produce a large, potential TR diversity. 

The expansion of the TRBV genes is certainly related to the complex pattern of gene duplications that led to the birth of a horse-specific germline repertoire, albeit through the emergence of a large numbers of non-functional TRBV genes. In fact, despite the large number of pseudogenes, the expression profile is broad, diversified, and similar to that of humans. 

Overall, these results provide insights into the horse immune system and offer many avenues for future investigation.

## Figures and Tables

**Figure 1 animals-14-02817-f001:**
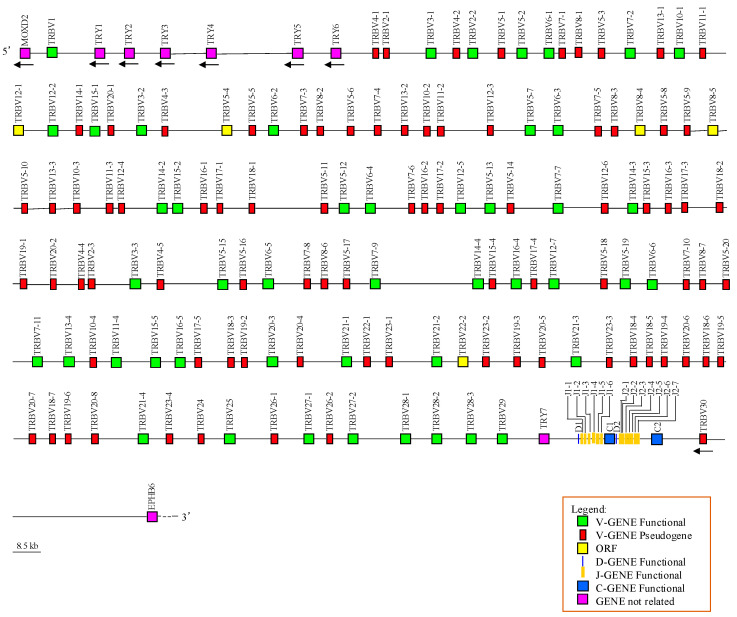
Schematic representation of the genomic organization of the horse TRB locus deduced from the EquCab3.0 genome assembly. The diagram shows the position of all related and unrelated TRB genes according to nomenclature. The boxes representing the genes are not to scale. The exons are not shown. The arrows indicate genes in reverse transcriptional orientation.

**Figure 2 animals-14-02817-f002:**
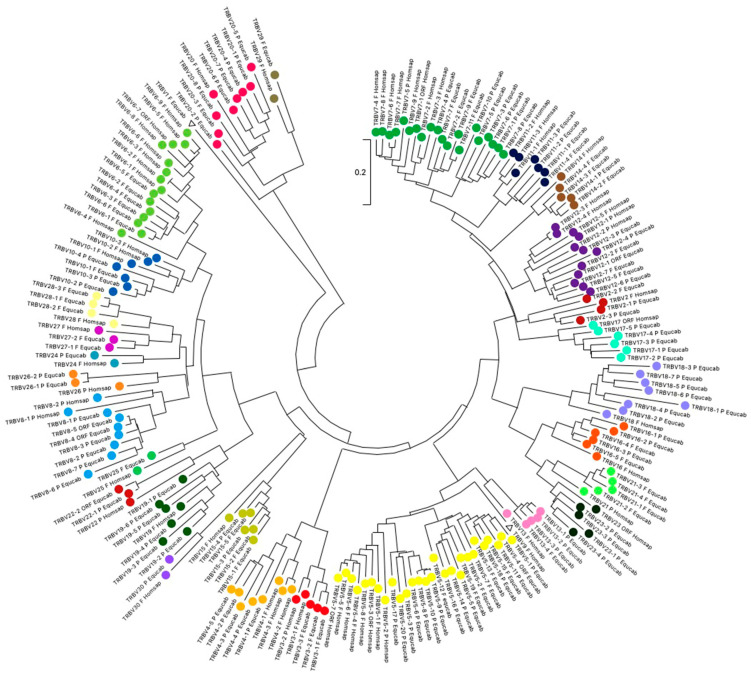
The neighbor-joining (NJ) tree inferred from the horse and human TRBV gene sequences. The evolutionary analyses were conducted in MEGA X [21,22]. The optimal tree is shown. The tree is drawn to scale, with branch lengths in the same units as those of the evolutionary distances used to infer the phylogenetic tree. The evolutionary distances were computed using the maximum composite likelihood method [24] and are in the units of the number of base substitutions per site. This analysis involved 200 nucleotide sequences. Codon positions included were 1st + 2nd + 3rd + noncoding. All ambiguous positions were removed for each sequence pair (pairwise deletion option). There was a total of 400 positions in the final dataset.

**Figure 3 animals-14-02817-f003:**
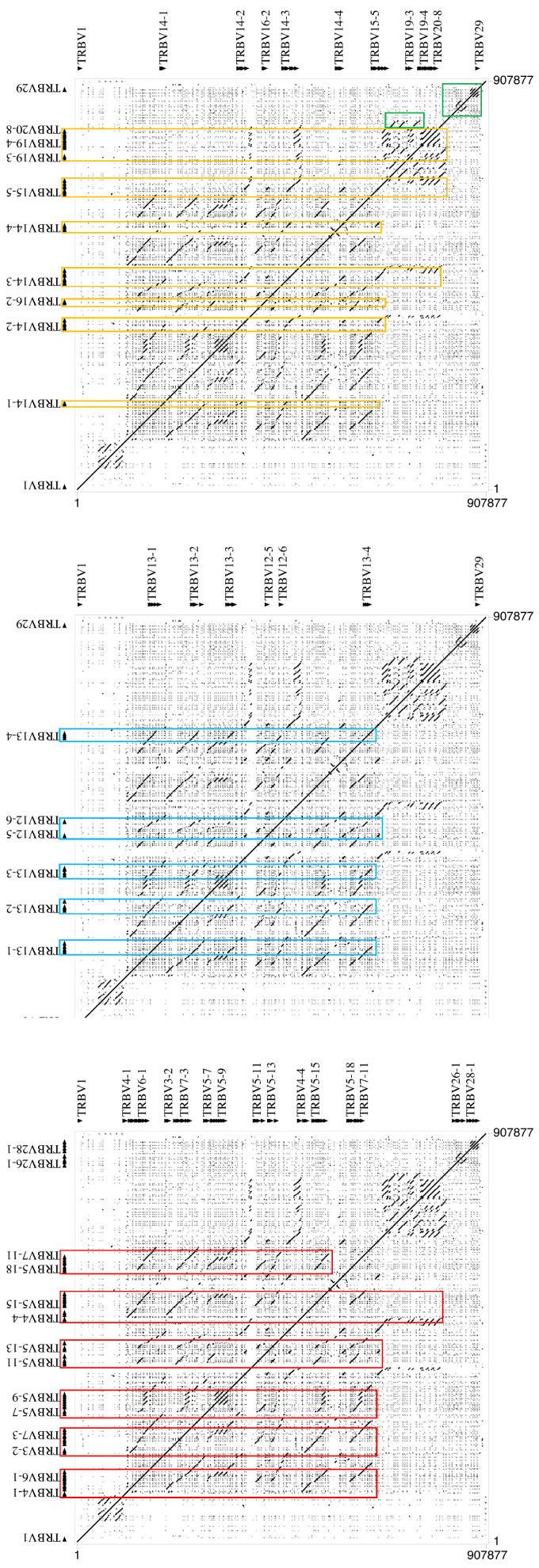
Dot plot of the horse TRBV cluster sequence against itself. With the exception of the main diagonal line for the match of each base with itself, dots and diagonal lines indicate internal homology units in the sequence. The matrix is tripled to show duplicated gene blocks derived from three distinct units (see text), each identified with a different color. Duplications involving the subgroups from TRBV2 to TRBV8 are boxed in red; duplications of the gene block involving the genes from TRBV10 to TRBV13 are highlighted in blue; the duplicated blocks of the genes from TRBV14 to TRBV20 are in yellow. The green boxes indicate the parallel lines of the duplications involving the TRBV21-TRBV22-TRBV23 and the TRBV26-TRBV27 gene units and the TRBV28 genes.

**Table 1 animals-14-02817-t001:** Correspondence between horse and human TRB genes as inferred from sequence analysis. The number of genes within each TRBV subgroup and the functionality (P, pseudogene; F, functional; ORF, open reading frame) are reported.

	Horse TRB				Human TRB			
Subgroups	No. of Genes	P	F	ORF	No. of Genes	P	F	ORF
TRBV1 *	1		1		1	1		
TRBV2	3	2	1		1		1	
TRBV3	3		3		2	1	1	
TRBV4	5	5			3		3	
TRBV5	20	13	6	1	8	1	5	2
TRBV6	6		6		8		7	1
TRBV7	11	7	4		9	1	7	1
TRBV8	7	5		2	2	2		
TRBV9	-				1		1	
TRBV10	4	3	1		3		3	
TRBV11	4	3	1		3		3	
TRBV12	7	3	3	1	5	2	3	
TRBV13	4	3	1		1		1	
TRBV14	4	1	3		1		1	
TRBV15	5	2	3		1		1	
TRBV16	5	3	2		1		1	
TRBV17	5	5			1			1
TRBV18	7	7			1		1	
TRBV19	6	6			1		1	
TRBV20	8	7	1		1		1	
TRBV21	4		4		1	1		
TRBV22	2	1		1	1	1		
TRBV23	4	4			1			1
TRBV24	1	1			1		1	
TRBV25	1		1		1		1	
TRBV26	2	2			1	1		
TRBV27	2		2		1		1	
TRBV28	3		3		1		1	
TRBV29	1		1		1		1	
TRBV30	1	1			1		1	
Total	136	84	47	5	64	11	47	6

* The correspondence is based only on the genomic position, both being the first gene within their own TRB locus.

**Table 2 animals-14-02817-t002:** Summary of the clonotypes based on TRBJ genes.

TRBJ Gene	N° of Clonotype	TRBD1 Gene	TRBD2 Gene	TRBD Gene (ND)	Mean CDR3 Length (AA)
TRBJ1-1	12	6	-	6	12.25 (range 9–16)
TRBJ1-2	25	14	2	9	12.04 (range 9–16)
TRBJ1-3	-	-	-	-	-
TRBJ1-4	4	2	-	2	12.75 (range 11–14)
TRBJ1-5	4	1	1	2	12.00 (range 11–13)
TRBJ1-6	19	9	1	9	12.57 (range 10–17)
TOTAL (Cluster1)	64	32	4	28	12.28 (range 9–17)

TRBJ2-1	42	9	22	11	12.69 (range 9–18)
TRBJ2-2	5	-	1	4	12.40 (range 10–14)
TRBJ2-3	21	6	7	8	12.90 (range 10–16)
TRBJ2-4	37	4	22	11	12.76 (range 10–16)
TRBJ2-5	12	1	8	3	12.17 (range 10–16)
TRBJ2-6	9	1	4	4	13.89 (range 12–17)
TRBJ2-7	22	-	13	9	12.54 (range 10–15)
TOTAL (Cluster2)	148	21	77	50	12.73 (range 9–18)
TOTAL (Cluster1 + Cluster2)	212	53	81	78	12.60 (range 9–18)

ND = Not determined.

**Table 3 animals-14-02817-t003:** Summary of the clonotypes based on TRBV genes. The total value of the analysis is highlighted in bold. The TRBV genes defined as non-functional in the genomic assembly are in italics.

Subgroups	N° of Clonotype	Resolved *
TRBV1		
TRBV2	2	V2-2 (2)
TRBV3	9	V3-3 (5)
TRBV4		
TRBV5	26	V5-2 (1) *V5-4* (1) V5-7 (4) V5-12 (3) V5-13 (4) V5-15 (6) V5-19 (1)
TRBV6	2	V6-6 (1)
TRBV7	7	V7-7 (4)
TRBV8	3	*V8-3* (2) *V8-5* (1)
TRBV9		
TRBV10		
TRBV11	1	V11-4 (1)
TRBV12	11	V12-2 (10) V12-7 (1)
TRBV13		
TRBV14	14	V14-3 (5) V14-4 (2)
TRBV15	6	V15-2 (2) V15-5 (3)
TRBV16		
TRBV17		
TRBV18		
TRBV19		
TRBV20	34	V20-3 (3) *V20-8* (26)
TRBV21	21	
TRBV22		
TRBV23		
TRBV24		
TRBV25	3	
TRBV26		
TRBV27	4	V27-1 (1)
TRBV28	17	
TRBV29	1	
TRBV30		
**TOTAL**	**161**	**93**

* Indicates the number of sequences for which a germline TRBV member has been conclusively assigned.

## Data Availability

Publicly available datasets were analyzed in this study. This data can be found here: [https://www.ncbi.nlm.nih.gov/; accession number: NC_009147].

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
