# Peer review of "A Comprehensive Analysis of the Genomic and Expressed Repertoire of the T-Cell Receptor Beta Chain in Equus caballus"

_animals, 2024, doi:10.3390/ani14192817_

Round 1

Reviewer 1 Report

Comments and Suggestions for Authors

The manuscript describes an analysis of the equine TRB locus and the repertoire of the various V, D, J and C genes therein. The manuscript is overall good and provides a clear and comprehensive description that meets the expressed aims of the study. There are a few issues that could be addressed by the authors:

Results/Discussions

·         From the locus map provided in Figure 1 I think that the equine TRBV1 is likely not the orthologue of the human TRBV1. Mouse (and cattle) have a TRBV gene (TRBV1 in mouse/TRBVX in cattle) that is located in this area of the TRB locus that has no orthologue in humans. I would suggest the authors download and check the nucleotide identity between the murine and equine TRBV1 to check this. May be worth considering extending this to a more complete comparison of the murine TRB gene repertoire with the equine locus?

·         Line 336-337 – could also reflect that TRBC sequences duplicated and diverged prior to speciation – not sure there is evidence of distinct roles of the TRBC isoforms.

·         Figure3 – need to have a more comprehensive legend description (e.g. the colour of the boxes in different panels of the Figure). Add a label to the different parts/panels of the figure.

·         TRB gene expression data – think the authors should make a clear statement that the repertoire examined for expression is relatively small and serves as a preliminary investigation of the expressed TRB gene repertoire and should be expanded in future studies to incorporate larger datasets to consolidate the observations made in the current study (comments for discussion?)

Minor errors/corrections:

1 – Line 40: change ‘juxtaposition’ to ‘(non-germline encoded) CDR3’

2 – Line 56/57: not sure I understand what the words ‘various nature’ refer to – please clarify

3 – Line 70: think the second parentheses should read (TRG) not (TRD).

4 – Line 73/74 – change ‘VD and DJ junctional sites’ to ‘V(D)J junctional sites’ to be comprehensive (TRA/TRG don’t have either VD or DJ junctions but instead VJ junctions)

5 – Line 22 and others: not sure ‘borne’ is the correct word here

6 – Line 233: references need correcting/inserting

7 – Line 242: change legend to reflect that arrows are only used to show genes in reverse orientation.

8 – Lines 254 and 255: change ‘not functional’ to ‘non-functional’

9 – Line 404: TRGJ genes? Should be TRBJ genes?

10 – Line 406: insert ‘more functionally’ in front of relevant

11 - Line 407: change ‘increase in use’ to ‘over-representation’

12 – Line 561: insert reference instead of ‘nostra ref dog’!!

Comments on the Quality of English Language

The English is generally of good quality but there are numerous minor grammatical errors throughout the manuscript (e.g. line 151 '....a germline gene is qualified as pseudogene.....' should read '....a germline gene is qualified as a pseudogene.....'

Author Response

Dear Reviewer,

Thank you very much for taking care of the reviewing process of our manuscript. We have reviewed and amended our manuscript according to your comments.

 Results/Discussions

  • From the locus map provided in Figure 1 I think that the equine TRBV1 is likely not the orthologue of the human TRBV1. Mouse (and cattle) have a TRBV gene (TRBV1 in mouse/TRBVX in cattle) that is located in this area of the TRB locus that has no orthologue in humans. I would suggest the authors download and check the nucleotide identity between the murine and equine TRBV1 to check this. May be worth considering extending this to a more complete comparison of the murine TRB gene repertoire with the equine locus?

Answer: Thanks for the relevant suggestion!

 We have verified the orthology of the horse TRBV1 with the mouse and bovine TRBV1 genes. Consequently, we have added the data to the results and inserted Figure S1.

According to the IMGT Committee (see Correspondence between Homo sapiens and Mus musculus TRBV subgroups nomenclature (IMGT-NC). Part I: Table and references DOI: 10.13140/RG.2.2.12205.05605)  the TRBV subgroups of newly annotated mammalian species are assigned based on the Homo sapiens subgroups. The only exception is the mouse TRBV subgroups which remained as originally assigned. Based on these indications, we have excluded the mouse genes from our analysis.

  • Line 336-337 – could also reflect that TRBC sequences duplicated and diverged prior to speciation – not sure there is evidence of distinct roles of the TRBC isoforms.

Answer: We agree with you! It is likely that the TRBC sequences have duplicated and diverged before speciation. However, there seems to be an evolutionary constraint (related to function?) that favors, in different species, the homogenization of the coding region of the two TRBC genes while maintaining the 3'UTRs different. This is what we are referring to.

  • Figure3 – need to have a more comprehensive legend description (e.g. the colour of the boxes in different panels of the Figure). Add a label to the different parts/panels of the figure.

Answer: we have added the description of each matrix in the figure legend.

  • TRB gene expression data – think the authors should make a clear statement that the repertoire examined for expression is relatively small and serves as a preliminary investigation of the expressed TRB gene repertoire and should be expanded in future studies to incorporate larger datasets to consolidate the observations made in the current study (comments for discussion?)

Answer: As you suggested, we have added the following clear statement about the clonotype analysis to the end of the Discussion section: “Furthermore, it must be said that the number of clonotype examined is relatively small so this can be considered a preliminary investigation of the expressed TRB gene repertoire in Equus caballus. Future studies should expand the same analysis incorporating larger datasets to consolidate the observations made in the current study.”

Minor errors/corrections:

1 – Line 40: change ‘juxtaposition’ to ‘(non-germline encoded) CDR3’

Answer: done

2 – Line 56/57: not sure I understand what the words ‘various nature’ refer to – please clarify

Answer:  we changed “peptides of various nature” to “different peptides”.

3 – Line 70: think the second parentheses should read (TRG) not (TRD).

Answer: It was corrected

4 – Line 73/74 – change ‘VD and DJ junctional sites’ to ‘V(D)J junctional sites’ to be comprehensive (TRA/TRG don’t have either VD or DJ junctions but instead VJ junctions)

Answer: done

5 – Line 22 and others: not sure ‘borne’ is the correct word here

Answer: The word is used according to the IMGT nomenclature (see https://www.imgt.org/IMGTrepertoire/LocusGenes/bornes/bornesTRB.html)

6 – Line 233: references need correcting/inserting

Answer: done

7 – Line 242: change legend to reflect that arrows are only used to show genes in reverse orientation.

Answer: done

8 – Lines 254 and 255: change ‘not functional’ to ‘non-functional’

Answer: done

9 – Line 404: TRGJ genes? Should be TRBJ genes?

Answer: It was corrected

10 – Line 406: insert ‘more functionally’ in front of relevant

Answer: done

11 - Line 407: change ‘increase in use’ to ‘over-representation’

Answer: done

12 – Line 561: insert reference instead of ‘nostra ref dog’!!

Answer: It was corrected

Comments on the Quality of English Language

The English is generally of good quality but there are numerous minor grammatical errors throughout the manuscript (e.g. line 151 '....a germline gene is qualified as pseudogene.....' should read '....a germline gene is qualified as a pseudogene.....'

Answer: A further editing of English language was done.

Once again, I would like to thank you for your recommendations.

Kind Regards

Reviewer 2 Report

Comments and Suggestions for Authors

This report contains the annotation of the locus encoding the horse T cell receptor b-chain (TRB), which spans approximately 1 Mb and is encompassing 136 TRBV,  2 TRBD, 13 TRBJ and 2 TRBC genes. An  analysis, using each equine TRBJ gene as a probe, provided insights into the usage of the TRBV, TRBD and TRBJ genes.

Comments:

Abstract: please clearly indicate that this a bioinformatic analyses from archived data sets and relys on human and equine sequences as well as human gene annotations. This should be said at the beginning of the Abstract and in the objectives of the Introduction.

Introduction: objectives may be improved. Please add and outline the reason of this study and what new information you expect to present without doing own experiments. The genomic data are already available, so this cannot be a new finding. You should say which genes are annotated and which ones are under ENSECAG- numbers and which are not present in the ref genome. 

So, you should explain what you do with these known sequences to come to a conclusion on a reliable gene annotation.

Line 110: species._This is due species >>  species. This is due species

Line 119-120: please show how do you did this. You need an accession number for this region.  If you apply Ensemble, you get synteny lists with other species. Why you did use NCBI and not both, NCBI and Ensemble. 

TRBV6 is downstream to EPHB6

TRPV6 (ENSECAG00000022404) 4:96073066-96126324
EPHB6 (ENSECAG00000007276) 4:96057953-96074952

Line 126: not clear. Which software did you to ensure ORF, exon-introns transitions, 3' UTR and 5' UTR regions, start of transcription, stop signal. Usually, this may be more complicated and 5' and 3' UTR are often difficult to determine. There is specific software to check these issues.

My general question to the authors: why did you not use software on gene prediction tools. There is a lot of software.

A point would be to crossvalidate the outcomes by the authors' approach with such tools. This may a limitation of the present study.

How can the reader be sure that all these data are reliable.

Author Response

Dear Reviewer,

Thank you very much for taking care of the reviewing process of our manuscript.

Our responses to your comments are below.

Abstract: please clearly indicate that this a bioinformatic analyses from archived data sets and relys on human and equine sequences as well as human gene annotations. This should be said at the beginning of the Abstract and in the objectives of the Introduction.

Introduction: objectives may be improved. Please add and outline the reason of this study and what new information you expect to present without doing own experiments. The genomic data are already available, so this cannot be a new finding. You should say which genes are annotated and which ones are under ENSECAG- numbers and which are not present in the ref genome. 

So, you should explain what you do with these known sequences to come to a conclusion on a reliable gene annotation.

Answer: Given the unconventional nature of the gene segments (V, D, J and C) that characterizes the TR loci, the analysis cannot be performed with common bioinformatics tools. We therefore conducted a genomic analysis with approaches strictly linked to these genes.

The annotation includes the correct classification of the genes according to an international nomenclature (IMGT-NC) that is not considered under ENSECAG-numbers.

We believe that in the short abstract and introduction it is clear where the analysis starts from and its objectives (… we report a comprehensive and consistent annotation of the locus encoding the horse T cell receptor b-chain (TRB), as inferred from the most recent genome assembly…; …we provide a comprehensive genomic and expression analysis of the TRB locus in Equus caballus ……. Based on the recently released horse genome…)

In any case, a sentence has been added at the end of the introduction for further clarification.

Line 110: species._This is due species >>  species. This is due species

Answer: done

Line 119-120: please show how do you did this.

Answer: This is described in M&M and Result sections.  The horse TRB sequence was identified in the genome assembly publicly available at NCBI, using the human one as reference. The human flanking genes (MOXD2 and EPHB6) were aligned by BLAST to define the sequence to extract.

You need an accession number for this region.  If you apply Ensemble, you get synteny lists with other species. Why you did use NCBI and not both, NCBI and Ensemble. 

TRBV6 is downstream to EPHB6

TRPV6 (ENSECAG00000022404)

4:96073066-96126324

EPHB6 (ENSECAG00000007276)

4:96057953-96074952

Answer: We believe that it is not strictly necessary to request an accession number of the region if in the article the authors describe the recovered sequence in detail. In fact, the exact positions of the genomic scaffold (NC_009147), available in the database, have been inserted, from which the entire horse sequence analyzed by us was retrieved. In addition, we also detailed in Table S1 the position of each identified gene in the sequence. Therefore, anyone can verify the reliability of the data. We used NCBI only as a database to retrieve the TRB sequence from the EquCab3.0 assembly. The same result would be obtained starting from Ensemble database.

The presence of the TRPV6 gene downstream to EPHB6 (as indicated in Ensemble) is not important for our analysis because it is outside the region of interest.

Line 126: not clear. Which software did you to ensure ORF, exon-introns transitions, 3' UTR and 5' UTR regions, start of transcription, stop signal. Usually, this may be more complicated and 5' and 3' UTR are often difficult to determine. There is specific software to check these issues.

My general question to the authors: why did you not use software on gene prediction tools. There is a lot of software.

A point would be to crossvalidate the outcomes by the authors' approach with such tools. This may a limitation of the present study.

How can the reader be sure that all these data are reliable.

Answer: Owing to their genetic complexity, these genes are challenging to analyze and classify. All horse TRB gene were analyzed according to IMGT gene nomenclature principles. “IMGT® (https://www.imgt.org/), the international ImMunoGeneTics® information system, established in 1989, is a high-quality integrated knowledge resource that manages sequences from genome to proteome, and structural data for immunoglobulins and T cell receptors in human and other jawed vertebrates. IMGT provides resources (database, tools, IMGT Repertoire as well as IG and TR genes and alleles reference sets) for jawed vertebrates for the analysis and understanding of immunogenetics.”

Based on this, all possible tools available at https://www.imgt.org/ were used to detect and delimitate the V, D, J and C genes along the retrieved genomic sequence. TRB genes were characterized and classified using alignments by BLAST as well as by implementing the IMGT unique numbering, and annotation rules of the IMGT Scientific chart, based on the IMGT-ONTOLOGY concepts for the genes and alleles functionalities.

Obviously, our proposed characterization and annotation of the horse TRB locus can be considered provisional, but as has already happened for other species, it represents an important reference sequence for future studies.

Once again, I would like to thank you.

Kind Regards

Round 2

Reviewer 2 Report

Comments and Suggestions for Authors

The authors made amendments to improve in their manuscript.

Some additional recommendations are given below:

Line 28: In this paper, we report a comprehensive and consistent annotation of the locus encoding the horse T cell receptor b-chain (TRB), as inferred from the most recent genome assembly. >>> In this paper, we report a comprehensive and consistent annotation of the locus encoding the b-chain of the equine T-cell receptor (TRB) as inferred from recent genome assembly using bioinformatics tools.

Line 173-174: For the horse phylogenetic analysis, we combined the nucleotide sequences of all V-REGION of the horse TRBV genes with the corresponding gene sequences of humans.  >>>For the phylogenetic analysis of the horse, we combined the nucleotide sequences of all V-REGION of the TRBV genes of the horse with the corresponding gene sequences of humans.

Legend of Table 1: please explain the abbreviations used in Table 1.

Comment to ORF: why you did not use biofomartic tools to check for ORFs, splice sites, etc. Just using human orthologues may be ok or not ok. You did not check this.

Figure 3: legend of the left side  is upside down.

Line 603: this conclusion is not really based on broad bioinformatic approaches or experimental procedures.  The authors should make a cautionary note here, even if the authors use horse transcriptome data from an RNA-Seq.

Comments on the Quality of English Language

No comments

Author Response

Dear Reviewer,

As per your request, we have taken on board you comments and made the following changes:

Line 28: In this paper, we report a comprehensive and consistent annotation of the locus encoding the horse T cell receptor b-chain (TRB), as inferred from the most recent genome assembly. >>> In this paper, we report a comprehensive and consistent annotation of the locus encoding the b-chain of the equine T-cell receptor (TRB) as inferred from recent genome assembly using bioinformatics tools.

Answer: done

 Line 173-174: For the horse phylogenetic analysis, we combined the nucleotide sequences of all V-REGION of the horse TRBV genes with the corresponding gene sequences of humans.  >>>For the phylogenetic analysis of the horse, we combined the nucleotide sequences of all V-REGION of the TRBV genes of the horse with the corresponding gene sequences of humans.

Answer: done

Legend of Table 1: please explain the abbreviations used in Table 1.

Answer: done

Comment to ORF: why you did not use biofomartic tools to check for ORFs, splice sites, etc. Just using human orthologues may be ok or not ok. You did not check this.

Answer: the functionality (i.e. ORF genes) are checked according the IMGT Scientific chart at https://www.imgt.org/IMGTScientificChart/SequenceDescription/IMGTfunctionality.html. Therefore, we use a simply sequence analysis software (such as Strider) to search for possible defects at the crucial sites, such as splicing sites or recombination signals, in each obviously characterized gene sequence.

Figure 3: legend of the left side  is upside down.

Answer: The legend is now below Figure 3

 Line 603: this conclusion is not really based on broad bioinformatic approaches or experimental procedures.  The authors should make a cautionary note here, even if the authors use horse transcriptome data from an RNA-Seq.

Answer: we refer to “germline” (we added it in the text) functional genes found within the genomic assemblies (horse and human ones) analyzed so far.

Kind Regards